# Total Synthesis of Nicrophorusamide A and Structural Disproof of the Proposed Noursamycin A

**DOI:** 10.3390/molecules28217442

**Published:** 2023-11-06

**Authors:** Xiaoyun Liao, Shupeng Li, Yian Guo, Tao Ye

**Affiliations:** 1State Key Laboratory of Chemical Oncogenomics, Peking University Shenzhen Graduate School, University Town, Xili, Shenzhen 518055, China; 2101111288@stu.pku.edu.cn (X.L.); yianguo@pku.edu.cn (Y.G.); 2QianYan (Shenzhen) Pharmatech. Ltd., 5th Floor East, Building-3, Longcheng Industrial Park, Qinglin Roas West, Longgang District, Shenzhen 518172, China

**Keywords:** total synthesis, cyclopeptide, noursamycin A, nicrophorusamide A

## Abstract

Total synthesis of the proposed noursamycin A has been accomplished, which disproves the original structural assignments. The synthetic strategy described herein has also been employed in the first total synthesis of nicrophorusamide A, a cyclopeptide that is structurally related to noursamycin A.

## 1. Introduction

Natural products have been playing a pivotal role in drug discovery [1]. Central to unearthing the pharmaceutical potential of these naturally occurring substances is the correct assignment of their unique structures, which enables downstream scientific investigations such as structure–activity relationship (SAR) studies. Despite significant advances in modern analytical methods, the process of determining the structures of natural products with limited availability can still be error prone [2,3]. In addition, total synthesis of natural products is an essential approach to establishing their authentic architectures. We have been making contributions in this field for many years, and the total synthesis program carried out by our team has resulted in the structural reassignment [4,5] or disproof [6] of a wide range of biologically intriguing natural products. Herein, we describe the synthetic efforts toward two structurally related cyclohexapeptides, which resulted in the disproving of noursamycin A’s original assignment [7] and the confirmation of nicrophorusamide A’s original assignment [8].

Noursamycins A and B were isolated in 2018, and their polypeptidyl structures were elucidated via mass spectrometric analyses and NMR spectroscopic methods [7]. The configurations of individual amino acids were identified via GC-MS analysis. Notably, the authors tentatively assigned *β*-hydroxyasparagine as d-*threo*-*β*-hydroxyasparagine based on a number of complementary findings including the NOE analysis of the OH-Asn subunit, the structural similarity of noursamycin B to nicrophorusamide A [8] and curacomycin, [9] and a putative biosynthetic pathway involving epimerization at the *α* position of OH-Asn. Biologically, the antibacterial activity of noursamycins A and B against Gram-positive and Gram-negative bacteria is exemplified by their inhibitory activity against *Bacillus subtilis* with MIC values as low as 4 μg/mL and 32 μg/mL, respectively. The retrosynthetic blueprint for the more potent congener, noursamycin A (**1**), is depicted below (Figure 1). The cyclic hexapeptide was strategically dissected in a convergent manner, and noursamycin A was envisioned to be assembled via an amidation reaction involving tripeptides **2** and **3**. Further simplification of these peptidic segments led to 5-chloro-l-tryptophan, d-valine, d-phenylalanine, l*-allo*-isoleucine, l-ornithine, and d-*β*-hydroxyasparagine derivatives (**4**–**9**).

## 2. Results and Discussion

### 2.1. Total Synthesis of the Proposed Noursamycin A *(**1**)*

As shown in Figure 2, the synthesis of tripeptide 2 was started with the preparation of the nonproteinogenic amino acid *(2R,3S*)-*β*-hydroxyasparagine derivative. d-Diethyl tartrate was rapidly converted into Fmoc and cyclic acetal-protected acid **10** according to a strategy employing Chen’s facile and practical protocol [10]. This was then masked as its 2-(trimethylsilyl)ethyl (TMSE) ester. In the event, acid **10** was treated with oxalyl chloride and catalytic dimethylformamide to afford the intermediary acyl chloride, and subsequent esterification went smoothly upon the addition of 2-(trimethylsilyl)ethanol, to give rise to ester **11** in 84% yield. Selective removal of the acetal protecting group was realized by heating a solution of **11** in AcOH/H_2_O/THF (2:2:1) at 45 °C, releasing the seco acid **12**, which subsequently reacted with allyl bromide and diisopropylethylamine in DMF to afford allyl ester **4**. The ensuing peptide elongation was commenced with 1,8-diazabicyclo[5.4.0]undec-7-ene (DBU)-elicited Fmoc deprotection, and the liberated amine coupled with *N*_δ_-Boc-*N*_α_-Fmoc-l-ornithine (**5**) in the presence of 1-[bis(dimethylamino)-methylene]-1*H*-1,2,3-triazolo[4,5-*b*]pyridinium 3-oxid hexafluorophosphate (HATU) [11] and 1-hydroxy-7-azabenzotriazole (HOAt), forging the dipeptide **13** in 67% yield. In an analogous manner, the dipeptide was further elongated to tripeptide **2** after Fmoc-deprotection and coupling with Fmoc-l-*allo*-isoleucine (**6**).

In parallel, we embarked on the synthesis of tripeptide **3** comprising a 5-Cl-Trp derivative. Spurred by Arnold’s elegant and rapid entry into Trp analogues [12], we conducted the preparation of the chlorinated tryptophan via enzymatic catalysis. In the event, l-Ser underwent enzymatic coupling with 5-Cl-indole triggered by a TrpB variant (PfTrpB2B9) in the presence of the cofactor pyridoxal phosphate (PLP), delivering 5-Cl-l-Trp **15** in one step without tedious chemical manipulations (Figure 3). Subsequently, protection of the amino terminus in **15** was executed under the condition of *N*-[2-(trimethylsilyl)ethoxycarbonyloxy]succinimide (Teoc-OSu)/TEA to afford acid **9** in 96% yield. Alongside the chemoenzymatic route to 5-Cl-l-Trp derivative, *O*-allyl-d-Phe **7** underwent amidation with *N*-Boc-d-Val to afford dipeptide **16**. Further elongation of the peptide chain was realized in two steps, namely acidic cleavage of the Boc group and peptide coupling with acid **9**, which generated tripeptide **3** in 74% over two steps.

With two tripeptides in hand, the stage was now set for their assembly and elaboration into noursamycin A (Figure 4). In the event, the DBU-promoted Fmoc deprotection of **2** liberated the amine, and unmasking the allyl ester **3** in the presence of an aryl chloride was achieved without incident under the mediation of Pd(PPh_3_)_4_ and morpholine. The subsequent amidation forged the hexapeptide skeleton, delivering **17** in 55% yield over two steps. Desilylation was then pursued and realized upon treating **17** with TBAF in THF at 50 °C. The thus acquired amino acid went through investigation into head-to-tail macrocyclization, which was best performed by implementing the EDCI/DMAP/HOAt system, (Appendix A) furnishing the cyclopeptide **18** in 65% yield over two steps. Upon dissolving macrocyclic peptide **18** in a saturated methanolic ammonia solution at 0 °C, transamidation of the allyl ester readily occurred and gave rise to the corresponding amide, which underwent Boc-deprotection to yield noursamycin A (**1**). However, comparison of the NMR spectroscopic data of **1** with those reported for natural noursamycin A revealed that they were similar, but not identical to those reported for the natural product. Discrepancies in the spectral data of the synthetic noursamycin A and those reported for the natural sample have led to the conclusion that the proposed structure of noursamycin A is incorrect. In order to verify the stereochemical integrity of our synthetic strategy we set out to synthesize antibacterial nicrophorusamide A (**25**), [6] which is structurally close to noursamycin A.

### 2.2. Total Synthesis of Nicrophorusamide A *(**25**)*

The key structural differences between nicrophorusamide A and the proposed noursamycin A are (1) replacement of the d-phenylalanine with the d-leucine, and (2) the absolute configuration of the hydroxy-bearing center present in the *β*-hydroxyasparagine subunit in nicrophorusamide A being opposite to that of the proposed noursamycin (Figure 5). It is worth mentioning that the absolute configuration of *β*-hydroxyasparagine in nicrophorusamide A was established as (*2R*,*3R*) via chemical derivatization and co-injection experiments with authentic samples [8]. The structural resemblance between these two cyclopeptides implied that the route devised for the proposed noursamycin A would be highly applicable to the preparation of nicrophorusamide A. Indeed, the facile access of nicrophorusamide A was pursued, and proceeded without incident. The synthesis commenced with the Fmoc and cyclic acetal-protected acid **19**, which was derived from l-dimethyl tartrate [13]. A three-step manipulation of **19** provided the desired (*2R,3R*)-*β*-hydroxyasparagine derivative **20** suitably adorned with appropriate protecting groups. The nonproteinogenic amino acid **20** was then elaborated to tripeptide **21** in 39% yield by an identical strategy as described for tripeptide **2**. With the key tripeptide **21** in hand, we next turned our attention to the synthesis of the lower fragment **23**. Thus, *O*-allyl-d-Leu **22** was reacted with *N*-Boc-d-Val using HATU with HOAt to afford the corresponding dipeptide in 80% yield. Removal of the Boc group with TFA gave the corresponding amine TFA salt, which was coupled with 5-Cl-Trp derivative **9**, using HATU to give tripeptide **23** in 91% yield over two steps. After unmasking the amino terminus in **21** and the acid terminus in **23**, fragment unification mediated by HATU/HOAt was conducted to furnish linear hexapeptide **24** in 83% yield over two steps. The elaboration of linear precursor **24** into nicrophorusamide A (**25**) was readily achieved by following the same synthetic procedure as for noursamycin A, which entailed the following conversions: TBAF-triggered desilylation, macrolactamization using EDCI/HOAt/DMAP, penultimate aminolysis of the allyl ester, and removal of the Boc group. To our delight, the spectroscopic data (^1^H and ^13^C NMR) of the synthetic sample were in all respect identical to those recorded in the literature for the natural nicrophorusamide A [8]. In addition, a negative optical rotation, ${\left[\alpha \right]}_{D}^{25}$ −15.5 (*c* 0.2, MeOH), was in close accord with the previously reported value ${\left[\alpha \right]}_{D}^{25}$ −18 (*c* 0.2, MeOH), thereby providing further confirmation of the absolute stereochemistry of the natural product. This total synthesis has allowed us to conclude that natural nicrophorusamide A is represented by the structure **25**.

## 3. Materials and Methods

### 3.1. General Experimental Details

All reactions were conducted in flame-dried or oven-dried glassware under an atmosphere of dry nitrogen or argon. Oxygen and/or moisture-sensitive solids and liquids were transferred appropriately. Concentration of solutions in a vacuum was accomplished using a rotary evaporator fitted with a water aspirator. Residual solvents were removed under high vacuum conditions (0.1–0.2 mm Hg). All reaction solvents were purified before use. Tetrahydrofuran was distilled from sodium benzophenone. Toluene was distilled over molten sodium metal. Dichloromethane, dimethylformamide, 1,2-dimethoxyethane, and diisoproylethylamine were distilled from CaH_2_. Methanol was distilled from Mg/I_2_. Flash column chromatography was performed using the indicated solvents on E. Qingdao silica gel 60 (230–400 mesh ASTM). TLC was carried out using pre-coated sheets (Qingdao silica gel 60-F250, 0.2 mm). Compounds were visualized with UV light, iodine, p-anisaldehyde stain, ceric ammonium molybdate stain, or phosphomolybdic acid in EtOH.

### 3.2. Procedures and Analytical Description of Compounds

Ester (**11**): To a suspension of carboxylic acid **10** (0.50 g, 1.2 mmol) in DCM (15 mL) was added DMF (0.93 μL, 0.012 mmol), followed by dropwise addition of oxalyl chloride (0.15 mL, 1.8 mmol) at 0 °C. The solution was allowed to warm to room temperature and stirred for 1.5 h. After the reaction was completed, the solvent was removed in vacuo to give the acid chloride that was used in the next step without further purification.

To a solution of the crude acid chloride in DCM (20 mL) was added 2-(trimethylsilyl) ethanol (0.25 mL, 1.8 mmol) carefully dropwise. The solution was stirred overnight at room temperature before it was quenched with 1 M HCl (5 mL). The organic layer was separated, washed with brine (10 mL), dried over Na_2_SO_4_, filtered, and concentrated. The crude product was purified by column chromatography on silica gel, eluting with EtOAc/hexanes to afford the corresponding ester **11** in 84% yield (0.41 g, 1.0 mmol) as a white amorphous solid: R_f_ = 0.62 (ethyl acetate/hexanes = 1:2); ${\left[\alpha \right]}_{D}^{25}$ = −0.90 (*c* 1.0, MeOH); ^1^H NMR (Methanol-*d*4, 500 MHz) *δ* 7.77 (2H, d, *J* = 7.5 Hz), 7.66 (2H, dd, *J* = 10.3, 7.5 Hz), 7.37 (2H, t, *J* = 7.4 Hz), 7.29 (2H, t, *J* = 7.5 Hz), 4.87 (1H, d, *J* = 2.6 Hz), 4.82 (1H, d, *J* = 2.5 Hz), 4.40 (1H, dd, *J* = 10.6, 6.9 Hz), 4.34 (1H, dd, *J* = 10.6, 7.0 Hz), 4.30–4.25 (1H, m), 4.25–4.17 (2H, m), 1.53 (3H, s), 1.51 (3H, s), 1.01 (2H, t, *J* = 8.6 Hz), 0.02 (9H, s); ^13^C NMR (Methanol-*d*4, 126 MHz) *δ* 171.5, 169.3, 158.5, 145.2, 145.1, 142.6, 142.5, 128.8, 128.2, 128.2, 126.3, 126.2, 120.9, 112.8, 76.8, 68.3, 65.5, 56.6, 48.3, 26.8, 26.1, 18.1, −1.6; HRMS (ESI) calculated for C_27_H_33_O_7_NSiNa^+^ [M + Na]^+^ 534.1920, found 534.1918.

Ester (**4**): The ester **11** (0.50 g, 1.2 mmol) was dissolved in a mixture of AcOH/H_2_O/THF 2:2:1 (10 mL) and heated at 45 °C for 12 h. The solvent was evaporated and the residue was azeotropically dried by co-evaporation from toluene (3 × 5 mL). The crude material was used immediately in the next step without further purification.

To a solution of the above carboxylic acid in anhydrous DMF (10 mL) at 0 °C under an Ar atmosphere was added DIPEA (0.26 mL, 1.5 mmol) and allyl bromide (0.13 mL, 1.5 mmol). The mixture was stirred at ambient temperature for 2 h and then quenched with H_2_O (5 mL). The product was extracted with EtOAc (2 × 30 mL) and the combined organic phases were washed with brine (20 mL), dried over Na_2_SO_4_, filtered, and concentrated. The crude product was purified by column chromatography on silica gel, eluting with EtOAc/hexanes to afford the corresponding allyl ester **4** in 70% yield (0.43 g, 0.84 mmol) as a white amorphous solid: R_f_ = 0.50 (ethyl acetate/hexanes = 1:2); ${\left[\alpha \right]}_{D}^{25}$ = +1.80 (*c* 1.0, MeOH); ^1^H NMR (Methanol-*d*4, 500 MHz) *δ* 7.78 (2H, d, *J* = 7.3 Hz), 7.67 (2H, t, *J* = 6.9 Hz), 7.38 (2H, t, *J* = 7.4 Hz), 7.30 (2H, t, *J* = 7.3 Hz), 5.97 (1H, ddt, *J* = 16.7, 11.2, 5.8 Hz), 5.37 (1H, dt, *J* = 17.4, 1.6 Hz), 5.23 (1H, d, *J* = 10.5 Hz), 4.86 (1H, s), 4.81 (1H, d, *J* = 3.5 Hz), 4.68 (2H, d, *J* = 5.8 Hz), 4.50 (1H, d, *J* = 3.5 Hz), 4.40 (1H, dd, *J* = 10.4, 6.8 Hz), 4.35–4.28 (1H, m), 4.21 (3H, ddt, *J* = 13.1, 9.3, 4.8 Hz), 1.06–0.92 (2H, m), 0.03 (9H, s); ^13^C NMR (Methanol-*d*4, 126 MHz) *δ* 172.3, 170.4, 158.4, 145.2, 145.2, 142.6, 133.3, 128.8, 128.2, 126.3, 126.2, 120.9, 119.0, 72.9, 68.3, 67.1, 65.1, 58.7, 48.3, 18.2, −1.6; HRMS (ESI) calculated for C_27_H_33_O_7_NSiNa^+^ [M + Na]^+^ 534.1920, found 534.1918.

Dipeptide (**13**): To a stirred solution of allyl ester **4** (0.77 g, 1.07 mmol) in dichloromethane (20 mL) at 0 °C under an atmosphere of argon was added DBU (0.16 mL, 1.07 mmol). The reaction mixture was stirred for 1.5 h at 0 °C, at which point TLC analysis indicated that the starting material was consumed completely. Then, HOAt (0.22 g, 1.60 mmol) was added and the mixture was stirred for 5 min at 0 °C. Acid **5** (0.53 g, 1.18 mmol) and HATU (0.61 g, 1.60 mmol) were added, and the resulting homogeneous yellow solution was allowed to warm to room temperature and stirred overnight. The reaction mixture was cooled to 0 °C and quenched by the addition of H_2_O (10 mL) and extracted with EtOAc (2 × 30 mL). The combined organic layers were dried over Na_2_SO_4_, filtered, and concentrated. The crude product was purified by column chromatography on silica gel, eluting with EtOAc/hexanes to afford dipeptide **13** in 67% yield (0.52 g, 0.72 mmol) as a white amorphous solid: R_f_ = 0.54 (ethyl acetate/hexanes = 1:1); ${\left[\alpha \right]}_{D}^{25}$ = −4.90 (*c* 1.0, MeOH); ^1^H NMR (Methanol-*d*4, 500 MHz) *δ* 7.79 (2H, dd, *J* = 7.7, 2.9 Hz), 7.70–7.63 (2H, m), 7.38 (2H, td, *J* = 7.5, 2.2 Hz), 7.31 (2H, t, *J* = 7.6 Hz), 5.97 (1H, ddt, *J* = 17.3, 11.8, 6.3 Hz), 5.37 (1H, d, *J* = 17.2 Hz), 5.24 (1H, d, *J* = 10.5 Hz), 5.03 (1H, d, *J* = 3.3 Hz), 4.81 (1H, s), 4.68 (2H, d, *J* = 5.8 Hz), 4.47 (1H, d, *J* = 3.3 Hz), 4.38 (2H, d, *J* = 7.1 Hz), 4.20 (4H, dt, *J* = 25.3, 6.2 Hz), 3.05 (2H, d, *J* = 7.1 Hz), 1.82 (1H, h, *J* = 5.5 Hz), 1.65 (1H, q, *J* = 5.1 Hz), 1.57–1.48 (1H, m), 1.43 (9H, s), 0.98 (2H, t, *J* = 8.7 Hz), 0.01 (9H, s); 13C NMR (Methanol-d4, 126 MHz) δ 174.6, 172.2, 169.8, 158.5, 145.3, 145.1, 142.6, 142.5, 133.2, 128.8, 128.2, 126.2, 126.2, 120.9, 119.0, 79.9, 72.5, 68.0, 67.1, 65.2, 56.8, 56.0, 40.8, 30.3, 28.8, 27.3, 18.2, −1.7; HRMS (ESI) calculated for C_37_H_51_O_10_N_3_SiNa^+^ [M + Na]^+^ 748.3237, found 748.3236.

Tripeptide (**2**): To a stirred solution of dipeptide **13** (0.38 g, 0.53 mmol) in dichloromethane (0 mL) at 0 °C under an atmosphere of argon was added DBU (78.5 μL, 0.53 mmol) via syringe. The reaction mixture was stirred for 1.5 h at 0 °C. After the starting material was consumed completely as determined by TLC, HOAt (0.11 g, 0.80 mmol) was added and the mixture was stirred for 5 min at 0 °C. Acid **6** (0.19 g, 0.53 mmol) followed by HATU (0.30 g, 0.80 mmol) were added and the resulting homogeneous yellow solution was allowed to warm to room temperature. The reaction mixture was cooled to 0 °C and quenched by the addition of H_2_O (5 mL) and extracted with EtOAc (2 × 15 mL). The combined organic layers were dried over Na_2_SO_4_, filtered, and concentrated. The crude product was purified by column chromatography on silica gel, eluting with EtOAc/hexanes to afford tripeptide **2** in 67% yield (0.30 g, 0.36 mmol) as a white amorphous solid: R_f_ = 0.55 (ethyl acetate/hexanes = 2:1); ${\left[\alpha \right]}_{D}^{25}$ = −7.40 (*c* 1.0, MeOH); ^1^H NMR (Methanol-*d*4, 400 MHz) *δ* 7.80 (2H, d, *J* = 7.5 Hz), 7.68 (2H, t, *J* = 8.1 Hz), 7.39 (2H, t, *J* = 7.5 Hz), 7.31 (2H, td, *J* = 7.4, 1.2 Hz), 5.98 (1H, ddt, *J* = 16.4, 10.9, 5.7 Hz), 5.37 (1H, dt, *J* = 17.2, 1.6 Hz), 5.25 (1H, dq, *J* = 10.5, 1.4 Hz), 5.03 (1H, d, *J* = 3.3 Hz), 4.72–4.62 (2H, m), 4.51 (1H, dd, *J* = 8.9, 5.1 Hz), 4.47 (1H, d, *J* = 3.3 Hz), 4.44–4.34 (2H, m), 4.24 (1H, t, *J* = 6.9 Hz), 4.17 (3H, dt, *J* = 12.2, 4.5 Hz), 3.03 (2H, q, *J* = 6.6 Hz), 1.90 (2H, tt, *J* = 13.7, 6.9 Hz), 1.66 (1H, td, *J* = 8.9, 4.6 Hz), 1.58–1.45 (3H, m), 1.40 (9H, s), 1.25–1.17 (1H, m), 1.01–0.83 (8H, m), 0.02 (9H, s); ^13^C NMR (Methanol-*d*4, 101 MHz) *δ* 174.6, 173.8, 172.2, 169.7, 158.8, 158.5, 145.4, 145.1, 142.5, 133.2, 128.7, 128.1, 128.1, 126.2, 120.9, 118.9, 79.9, 72.5, 68.0, 67.1, 65.1, 60.4, 56.8, 54.0, 48.4, 40.7, 38.2, 30.2, 28.7, 27.2, 18.2, 15.0, 11.9, −1.7; HRMS (ESI) calculated for C_43_H_62_O_11_N_4_SiNa^+^ [M + Na]^+^ 861.4077, found 861.4077.

Dipeptide (**16**): To a stirred solution of *N*-Boc-d-valine (1.24 g, 5.71 mmol) and d-phenylalanine allyl ester **7** (1.23 g, 6.0 mmol) in DMF (50 mL) at 0 °C under an atmosphere of argon was added DIPEA (2.98 mL, 17.13 mmol) and HOAt (1.17 g, 8.57 mmol), and the mixture was stirred for 5 min. HATU (3.25 g, 8.57 mmol) was then added, and the resulting homogeneous reaction mixture was allowed to warm to room temperature. The reaction mixture was cooled to 0 °C, quenched by the addition of H_2_O (10 mL) and extracted with EtOAc (2 × 50 mL). The combined organic layers were washed with aqueous 1 N HCl (2 × 30 mL), saturated aqueous NaHCO_3_ (2 × 30 mL), H_2_O (30 mL), and brine (30 mL) and dried over Na_2_SO_4_, filtered, and concentrated. The crude product was purified by column chromatography on silica gel, eluting with EtOAc/hexanes to afford dipeptide **16** in 84% yield (1.93 g, 4.80 mmol) as a white amorphous solid: R_f_ = 0.55 (ethyl acetate/hexanes = 1:2); ${\left[\alpha \right]}_{D}^{25}$ = +19.30 (*c* 1.0, MeOH); ^1^H NMR (Methanol-*d*4, 500 MHz) *δ* 7.29–7.16 (5H, m), 5.85 (1H, ddt, *J* = 16.4, 10.9, 5.7 Hz), 5.30–5.24 (1H, m), 5.19 (1H, dd, *J* = 10.6, 1.5 Hz), 4.71 (1H, dd, *J* = 8.5, 6.0 Hz), 4.56 (2H, dq, *J* = 5.5, 1.8 Hz), 3.86 (1H, d, *J* = 7.2 Hz), 3.15 (1H, dd, *J* = 13.9, 6.1 Hz), 3.02 (1H, dd, *J* = 13.8, 8.4 Hz), 1.94 (1H, dt, *J* = 13.9, 7.0 Hz), 1.43 (9H, s), 0.87 (6H, dd, *J* = 10.2, 6.8 Hz); ^13^C NMR (Methanol-*d*4, 126 MHz) *δ* 174.3, 172.4, 157.8, 138.0, 133.2, 130.3, 129.5, 127.9, 118.9, 80.5, 66.8, 61.4, 55.1, 38.5, 32.1, 28.7, 19.7, 18.5; HRMS (ESI) calculated for C_22_H_32_O_5_N_2_Na^+^ [M + Na]^+^ 427.2205, found 427.2203.

l-5-chloro-tryptophan hydrochloride (**15**): In a 100 mL round-bottomed flask, 5-Cl-indole (773.11 mg, 5.10 mmol), l-serine **14** (589.55 mg, 5.61 mmol) and pyridoxal phosphate (26.52 mg, 0.10 mmol) were suspended in DMSO (1.0 mL) and 50 mM KPi buffer (20 mL). The vial was sealed and then placed in a water bath that had been equilibrated to 70 °C. After 1 min, the enzyme was added. The reaction was kept at 70 °C. After 12 h, the reaction mixture was cooled in an ice bath for 90 min. The precipitate was collected by filtration and washed with ethyl acetate (2 × 20 mL) and ice water (2 × 20 mL). The precipitate was dissolved in a mixture of 1 M hydrochloric acid and acetonitrile (1:1), filtered, and the filtrate was concentrated in vacuo to afford l-5-chloro-tryptophan hydrochloride **15** in 50% yield (698.78 mg, 2.55 mmol) as a white amorphous solid: ${\left[\alpha \right]}_{D}^{25}$ = +6.70 (*c* 1.0, MeOH); ^1^H NMR (400 MHz, DMSO-*d*6) *δ* 11.38 (d, *J* = 2.7 Hz, 1H), 8.43 (s, 3H), 7.64 (d, *J* = 2.0 Hz, 1H), 7.39 (d, *J* = 8.6 Hz, 1H), 7.34 (d, *J* = 2.4 Hz, 1H), 7.07 (dd, *J* = 8.6, 2.1 Hz, 1H), 4.10 (t, *J* = 6.0 Hz, 1H), 3.28 (d, *J* = 5.9 Hz, 2H); ^13^C NMR (101 MHz, DMSO) *δ* 170.8, 134.8, 128.4, 127.0, 123.5, 121.2, 117.8, 113.2, 106.8, 52.7, 25.8; HRMS (ESI) calculated for C_11_H_11_ClN_2_O_2_H^+^ [M + H]^+^ 239.0585, found 239.0582.

Teoc-protected amino acid (**9**): A mixture of Teoc-OSu (1.04 g, 4.0 mmol), **15** (0.95 g, 4.0 mmol), and Et_3_N (0.17 mL, 1.2 mmol) in 1:1 H_2_O-dioxane (40 mL) was stirred for 16 h before it was diluted with 1 N KHSO_4_ (10 mL). The aqueous layer was extracted with ether (4 × 50 mL). The combined organic layers were washed with water (2 × 20 mL), dried over Na_2_SO_4_, filtered, and concentrated. The crude product was purified by column chromatography on silica gel, eluting with EtOAc/hexanes, providing the Teoc-protected amino acid **9** in 96% yield (1.47 g, 3.84 mmol) as a white amorphous solid: R_f_ = 0.55 (ethyl acetate/hexanes = 1:1); ${\left[\alpha \right]}_{D}^{25}$ = −3.50 (*c* 1.0, MeOH); ^1^H NMR (Methanol-*d*4, 500 MHz) *δ* 7.54 (1H, d, *J* = 2.0 Hz), 7.27 (1H, d, *J* = 8.6 Hz), 7.13 (1H, s), 7.04 (1H, dd, *J* = 8.5, 2.0 Hz), 4.47 (1H, dd, *J* = 7.8, 5.2 Hz), 4.15–4.00 (2H, m), 3.29 (1H, d, *J* = 5.0 Hz), 3.12 (1H, dd, *J* = 14.7, 7.8 Hz), 0.93 (2H, t, *J* = 8.5 Hz), −0.00 (9H, s); ^13^C NMR (Methanol-*d*4, 126 MHz) *δ* 175.5, 158.6, 136.3, 130.1, 126.6, 126.2, 125.6, 122.5, 118.8, 113.5, 111.0, 64.2, 56.2, 28.5, 18.6, −1.5; HRMS (ESI) calculated for C_17_H_23_O_4_N_2_ClSiNa^+^ [M + Na]^+^ 405.1008, found 405.1008.

Tripeptide (**3**): To a solution of compound **16** (40.42 mg, 0.1 mmol) in dry DCM (2 mL) at 0 °C was added Trifluoroacetic acid (1 mL). The reaction mixture was stirred at room temperature for 2 h. Volatiles were removed in vacuo and the residue was dried under high vacuum conditions for 2 h to afford the corresponding amine, which was used in the next step without further purification.

To a solution of amino acid **9** (41.41 mg, 0.1 mmol) in DCM (5 mL) at 0 °C were added DIPEA (69.68 μL, 0.4 mmol) and HOAt (20.42 mg, 0.15 mmol), and the mixture was stirred for 5 min. HATU (57.03 mg, 0.15 mmol) was added and the resulting homogeneous reaction mixture was allowed to warm to room temperature. The reaction mixture was cooled to 0 °C and quenched by the addition of H_2_O (2 mL) and extracted with EtOAc (3 × 10 mL). The combined organic layers were washed with aqueous 1 N HCl (2 × 10 mL), saturated aqueous NaHCO_3_ (2 × 10 mL), H_2_O (10 mL), and brine (10 mL), dried over Na_2_SO_4_, filtered, and concentrated. The crude product was purified by column chromatography on silica gel, eluting with EtOAc/hexanes to afford tripeptide **3** in 74% yield over two steps (0.049 g, 0.074 mmol) as a white amorphous solid: R_f_ = 0.56 (ethyl acetate/hexanes = 2:1); ${\left[\alpha \right]}_{D}^{25}$ = +13.20 (*c* 1.0, MeOH); ^1^H NMR (Methanol-*d*4, 500 MHz) *δ* 7.61–7.57 (1H, m), 7.30–7.17 (5H, m), 7.20–7.13 (1H, m), 7.12 (1H, s), 7.04 (1H, dd, *J* = 8.7, 2.1 Hz), 5.85 (1H, ddt, *J* = 16.4, 10.9, 5.7 Hz), 5.26 (1H, dd, *J* = 17.2, 1.7 Hz), 5.18 (1H, dd, *J* = 10.5, 1.5 Hz), 4.66 (1H, dt, *J* = 9.4, 5.6 Hz), 4.55 (2H, dd, *J* = 5.6, 1.4 Hz), 4.42 (1H, t, *J* = 7.3 Hz), 4.20–4.00 (3H, m), 3.21–3.13 (2H, m), 3.03 (2H, dt, *J* = 14.2, 8.9 Hz), 1.99–1.86 (1H, m, *J* = 6.8 Hz), 0.96 (2H, t, *J* = 8.6 Hz), 0.51 (6H, dd, *J* = 11.9, 6.8 Hz), 0.00 (9H, s); ^13^C NMR (Methanol-*d*4, 126 MHz) *δ* 174.9, 173.5, 172.4, 158.7, 138.2, 136.4, 133.2, 130.3, 129.9, 129.5, 127.8, 126.5, 125.6, 122.6, 119.0, 118.8, 113.5, 110.7, 66.8, 64.3, 59.8, 57.5, 55.3, 38.2, 31.0, 28.6, 19.3, 18.6, 17.7, −1.5; HRMS (ESI) calculated for C_34_H_45_O_6_N_4_ClSiNa^+^ [M + Na]^+^ 691.2691, found 691.2689.

Hexapeptide (**17**): To a solution of allyl ester **3** (80.19 mg, 0.12 mmol) in dry THF (5 mL) was added Pd(PPh_3_)_4_ (6.93 mg, 0.006 mmol), followed by the dropwise addition of morpholine (0.1 mL, 1.20 mmol). The reaction mixture was stirred for 45 min at ambient temperature, and then concentrated. The residue was dissolved in DCM (30 mL) and washed with 1 N HCl (10 mL) and H_2_O (10 mL). The organic layer was dried with anhydrous Na_2_SO_4_ and concentrated. The crude acid intermediate **S1** was obtained as an oil which was used in the next step without further purification. To a solution of Tripeptide **2** (100.61 mg, 0.12 mmol) in dichloromethane (10 mL) at 0 °C was added DBU (80.88 μL, 0.12 mmol), and stirred for 1.5 h. After the starting material was consumed completely as determined by TLC, HOAt (24.50 mg, 0.18 mmol) was added and the mixture was stirred for 5 min. Then, **S1** (78.97 mg, 0.12 mmol) and HATU (68.44 mg, 0.18 mmol) were added, and the resulting homogeneous yellow solution was allowed to warm to room temperature. The reaction mixture was cooled to 0 °C, quenched by the addition of H_2_O (2 mL), and extracted with EtOAc (2 × 10 mL). The combined organic layers were dried over Na_2_SO_4_, filtered, and concentrated. The crude product was purified by column chromatography on silica gel, eluting with EtOAc/hexanes to afford hexapeptide **17** in 55% yield over two steps (0.081 g, 0.066 mmol) as a white amorphous solid: R_f_ = 0.31 (methanol/dichloromethane = 1: 20); ${\left[\alpha \right]}_{D}^{25}$ = −13.80 (*c* 1.0, MeOH:DCM = 1:1); ^1^H NMR (DMSO-*d*6, 500 MHz) *δ* 11.00 (1H, d, *J* = 2.4 Hz), 8.17 (2H, t, *J* = 10.0 Hz), 7.99 (2H, dd, *J* = 12.7, 8.4 Hz), 7.83 (1H, d, *J* = 8.9 Hz), 7.70 (1H, s), 7.32 (1H, d, *J* = 8.7 Hz), 7.25–7.19 (5H, m), 7.14 (2H, dd, *J* = 12.9, 7.7 Hz), 7.03 (1H, dd, *J* = 8.6, 2.0 Hz), 6.74 (1H, t, *J* = 5.8 Hz), 6.05 (1H, s), 5.93 (1H, ddt, *J* = 16.2, 10.7, 5.6 Hz), 5.35 (1H, dt, *J* = 17.3, 1.7 Hz), 5.23 (1H, dd, *J* = 10.4, 1.6 Hz), 4.85 (1H, dd, *J* = 8.6, 3.8 Hz), 4.71–4.62 (1H, m), 4.61 (2H, dd, *J* = 5.4, 2.8 Hz), 4.40 (1H, td, *J* = 8.5, 5.4 Hz), 4.34 (2H, td, *J* = 8.8, 5.1 Hz), 4.27 (1H, d, *J* = 3.8 Hz), 4.16 (1H, t, *J* = 7.7 Hz), 4.13–4.01 (2H, m), 3.94 (2H, t, *J* = 8.5 Hz), 3.02 (1H, dd, *J* = 14.4, 5.1 Hz), 2.96 (1H, dd, *J* = 13.7, 5.8 Hz), 2.93–2.78 (4H, m), 2.00–1.90 (1H, m), 1.76 (1H, p, *J* = 6.6 Hz), 1.68–1.59 (1H, m), 1.50 (1H, d, *J* = 10.5 Hz), 1.36 (11H, s), 1.15 (1H, dd, *J* = 13.9, 6.8 Hz), 0.94–0.88 (3H, m), 0.84 (2H, td, *J* = 7.9, 2.8 Hz), 0.77 (3H, t, *J* = 7.3 Hz), 0.68 (11H, dt, *J* = 10.3, 3.9 Hz), −0.02 (18H, d, *J* = 26.5 Hz); ^13^C NMR (DMSO-*d*6, 126 MHz) δ 172.0, 171.6, 170.9, 170.8, 170.6, 170.4, 168.5, 156.1, 155.6, 137.5, 134.5, 132.3, 129.1, 128.5, 128.0, 126.1, 125.9, 123.0, 120.7, 118.0, 117.9, 112.7, 110.1, 77.4, 71.3, 65.0, 63.0, 61.8, 57.4, 55.5, 54.9, 54.2, 52.0, 37.9, 37.0, 31.3, 30.3, 29.3, 29.0, 28.3, 27.8, 25.9, 25.6, 19.1, 17.6, 17.3, 16.8, 14.2, 11.6, −1.6, −1.6; HRMS (ESI) calculated for C_59_H_91_N_8_O_14_Si_2_ClNa^+^ [M + Na]^+^ 1249.5779, found 1249.5774.

Hexapeptide (**18**): To a solution of the hexapeptide **17** (355.71 mg, 0.29 mmol) in THF (5 mL) was added 1.0 M TBAF in THF (2.9 mL, 2.9 mmol). The reaction mixture was stirred at 50 °C for 1 h and then cooled to 0 °C and quenched by the addition of a saturated aqueous solution of NH_4_Cl (5 mL). The reaction mixture was transferred to a separatory funnel and extracted with EtOAc (3 × 30 mL). The combined organic layers were dried over Na_2_SO_4_, filtered, and concentrated. The crude product was used in the next step without further purification.

To a solution of the above crude product in a mixture of dichloromethane (280 mL) and DMF (15 mL) were sequentially added DMAP (7.09 mg, 0.058 mmol), HOAt (78.94 mg, 0.58 mmol), and EDCI (111.19 mg, 0.58 mmol). The reaction mixture was stirred at 30 °C for 48 h and then cooled to 0 °C and quenched by the addition of a saturated aqueous solution of NH_4_Cl (20 mL). The mixture was transferred to a separatory funnel where the aqueous layer was extracted with DCM (3 × 30 mL). The combined organic layers were washed with H_2_O (2 × 50 mL) and brine (50 mL), dried over Na_2_SO_4_, and concentrated. The residue was purified by column chromatography on silica gel, eluting with MeOH/DCM to afford hexapeptide **18** in 65% yield (0.18 g, 0.19 mmol) as a white amorphous solid: R_f_ = 0.33 (methanol/dichloromethane = 1:20); ${\left[\alpha \right]}_{D}^{25}$ = +6.60 (*c* 1.0, MeOH:DCM = 1:1); ^1^H NMR (DMSO-*d*6, 400 MHz) *δ* 11.02 (1H, d, *J* = 2.5 Hz), 8.17 (1H, d, *J* = 7.2 Hz), 7.87 (2H, dd, *J* = 8.0, 4.2 Hz), 7.81 (1H, d, *J* = 7.2 Hz), 7.73 (1H, dd, *J* = 15.7, 8.6 Hz), 7.54 (1H, d, *J* = 2.0 Hz), 7.46 (1H, d, *J* = 6.7 Hz), 7.34–7.08 (7H, m), 7.03 (1H, dd, *J* = 8.6, 2.1 Hz), 6.77 (1H, t, *J* = 5.8 Hz), 6.05 (1H, dd, *J* = 6.4, 2.0 Hz), 6.01–5.83 (1H, m), 5.32 (1H, dq, *J* = 17.3, 1.7 Hz), 5.20 (1H, dq, *J* = 10.5, 1.5 Hz), 4.66 (2H, q, *J* = 7.0, 6.5 Hz), 4.64–4.42 (3H, m), 4.27 (1H, td, *J* = 7.9, 7.3, 3.0 Hz), 4.17–4.09 (1H, m), 3.96–3.88 (1H, m), 3.88–3.78 (1H, m), 3.11 (1H, dd, *J* = 14.2, 7.9 Hz), 3.04–2.80 (5H, m), 1.90–1.66 (2H, m), 1.50 (1H, d, *J* = 8.2 Hz), 1.35 (10H, s), 1.24 (1H, d, *J* = 10.1 Hz), 1.01–0.78 (3H, m), 0.75–0.64 (6H, m), 0.55 (6H, dt, *J* = 17.9, 6.5 Hz); ^13^C NMR (DMSO-*d*6, 101 MHz) *δ* 171.9, 171.4, 171.1, 171.0, 170.9, 170.7, 168.6, 155.5, 137.1, 137.0, 134.5, 132.4, 129.1, 128.3, 128.1, 126.3, 125.7, 123.0, 120.7, 117.9, 117.5, 113.3, 112.8, 109.5, 77.3, 70.6, 65.0, 61.8, 59.5, 57.6, 54.9, 54.5, 53.7, 52.6, 37.9, 35.8, 29.5, 28.2, 28.1, 26.7, 26.3, 25.4, 18.9, 17.9, 14.2, 11.5; HRMS (ESI) calculated for C_48_H_65_O_11_N_8_ClNa^+^ [M + Na]^+^ 987.4351, found 987.4354.

Noursamycin A (**1**): Cyclic hexapeptide **18** (28.93 mg, 0.03 mmol) was dissolved in a saturated methanolic NH_3_ solution (10 mL) and stirred at 0 °C. After the starting material was consumed completely as determined by TLC, the reaction mixture was concentrated. The residue was azeotropically dried by concentration from methanol (2 × 10 mL) to remove ammonia from the product. The residue was used in the next step without further purification.

Trifluoroacetic acid (1 mL) was added to a solution of the above amide (27.70 mg, 0.03 mmol) in dry DCM (3 mL) at 0 °C. The reaction mixture was stirred for 0.5 h at rt. Volatiles were removed in vacuo and the residue was purified by column chromatography on silica gel, eluting with MeOH/DCM to afford **1** in 58% yield over two steps (14.33 mg, 0.017 mmol) as a white amorphous solid: R_f_ = 0.23 (methanol/dichloromethane = 1:2); ${\left[\alpha \right]}_{D}^{25}$ = +11.40 (*c* 1.0, MeOH); IR (MeOH) 3472, 3385, 2967, 2073, 1632, 1545, 1205, 1053, 1016 cm^−1^; ^1^H NMR (DMSO-*d*6, 500 MHz) *δ* 11.02 (1H, s), 8.09 (1H, d, *J* = 8.2 Hz), 7.97 (2H, d, *J* = 8.0 Hz), 7.81 (1H, d, *J* = 7.5 Hz), 7.76 (2H, d, *J* = 6.0 Hz), 7.76–7.69 (2H, m), 7.59 (1H, s), 7.48 (1H, s), 7.33 (2H, dd, *J* = 19.4, 8.1 Hz), 7.25 (2H, t, *J* = 7.5 Hz), 7.16 (3H, dd, *J* = 12.1, 7.2 Hz), 7.12 (1H, s), 7.08–7.00 (1H, m), 6.99 (1H, s), 4.79 (1H, q, *J* = 7.3 Hz), 4.56 (1H, q, *J* = 7.9 Hz), 4.50 (2H, s), 4.18 (1H, q, *J* = 8.0, 7.6 Hz), 3.89–3.79 (2H, m), 3.31 (1H, dd, *J* = 14.9, 5.3 Hz), 3.01–2.90 (2H, m), 2.73 (3H, dd, *J* = 15.2, 9.0 Hz), 2.01–1.92 (1H, m), 1.86 (1H, hept, *J* = 6.9 Hz), 1.72 (1H, h, *J* = 6.6 Hz), 1.55 (1H, dd, *J* = 15.5, 9.0 Hz), 1.51–1.42 (2H, m), 0.91 (2H, dtt, *J* = 29.3, 15.5, 7.4 Hz), 0.74–0.68 (9H, m), 0.65 (3H, d, *J* = 6.6 Hz); ^13^C NMR (DMSO-*d*6, 126 MHz) *δ* 174.2, 172.1, 171.4, 171.1, 170.7, 170.6, 169.0, 137.2, 134.5, 129.3, 128.3, 128.1, 126.3, 125.6, 123.0, 120.7, 117.3, 112.8, 110.1, 70.6, 60.6, 58.2, 56.6, 53.9, 52.6, 52.4, 38.5, 35.8, 29.7, 27.7, 26.1, 25.4, 24.2, 19.1, 18.8, 14.3, 11.6; HRMS (ESI) calculated for C_40_H_55_N_9_O_8_Cl^+^ [M + H]^+^ 824.3857, found 824.3857.

Ester (**S2**): To a suspension of carboxylic acid **19** (2.76 g, 6.7 mmol) in DCM (40 mL), oxalyl chloride (0.68 mL, 8.0 mmol) was added dropwise at 0 °C, followed by the addition of DMF (5.67 μL, 0.067 mmol). The solution was allowed to warm to room temperature and stirred continuously for 2 h. After the reaction was completed, the solvent was removed in vacuo to give the acid chloride that was used in the next step without additional purification.

To a solution of the crude acid chloride in DCM (20 mL) was added 2-(trimethylsilyl) ethanol (1.05 mL, 7.4 mmol) dropwise. The mixture was stirred overnight at room temperature before it was quenched with 1 M HCl (5 mL). The organic layer was separated, washed with brine (10 mL), dried over Na_2_SO_4_, filtered, and concentrated. The crude product was purified by column chromatography on silica gel, eluting with EtOAc/hexanes to afford the corresponding ester **S2** in 79% yield (2.71 g, 5.3 mmol) as a white amorphous solid: R_f_ = 0.66 (ethyl acetate/hexanes = 1:2); ${\left[\alpha \right]}_{D}^{25}$ = +11.10 (*c* 1.0, MeOH); ^1^H NMR (Methanol-*d*4, 500 MHz) *δ* 7.79 (2H, d, *J* = 7.5 Hz), 7.66 (2H, d, *J* = 7.5 Hz), 7.39 (2H, t, *J* = 7.3 Hz), 7.30 (2H, tt, *J* = 7.5, 1.5 Hz), 5.10 (1H, d, *J* = 2.4 Hz), 4.40–4.31 (2H, m), 4.33–4.26 (2H, m), 4.24 (1H, t, *J* = 7.1 Hz), 1.66 (3H, s), 1.57 (3H, s), 1.29 (1H, d, *J* = 6.9 Hz), 1.09–1.01 (2H, m), 0.05 (9H, s); ^13^C NMR (Methanol-*d*4, 126 MHz) *δ* 171.7, 170.3, 158.6, 145.2, 145.1, 142.6, 128.8, 128.2, 126.3, 126.3, 120.9, 112.9, 76.4, 68.5, 65.6, 55.6, 48.3, 26.7, 26.0, 18.2, −1.5; HRMS (ESI) calculated for C_27_H_33_O_7_NSiNa^+^ [M + Na]^+^ 534.1920, found 534.1918.

Ester (**20**): The ester **S2** (2.45 g, 4.8 mmol) was dissolved in a mixture of AcOH/H_2_O/THF 2:2:1 (50 mL) and heated to 45 °C for 12 h. The solvent was evaporated, and the residue was azeotropically dried by concentration from toluene (3 × 5 mL). The crude material was submitted to the following step without purification.

The above carboxylic acid was dissolved in anhydrous DMF (30 mL) and cooled to 0 °C under an Ar atmosphere. Then, DIPEA (1.35 mL, 7.2 mmol) and allyl bromide (0.62 mL, 7.2 mmol) were added sequentially. The mixture was left to stir at ambient temperature for 2 h. The reaction was quenched with H_2_O (10 mL) and the product was extracted with EtOAc (2 × 50 mL). The combined organic phases were washed with brine (20 mL), dried (Na_2_SO_4_), filtered, and concentrated. The crude product was purified by column chromatography on silica gel, eluting with EtOAc/hexanes to afford the corresponding allyl ester **20** in 72% yield over two steps (1.77 g, 3.5 mmol) as a white amorphous solid: R_f_ = 0.43 (ethyl acetate/hexanes = 1:2); ${\left[\alpha \right]}_{D}^{25}$ = +4.40 (*c* 1.0, MeOH); ^1^H NMR (Methanol-*d*4, 500 MHz) *δ* 7.78 (2H, d, *J* = 7.5 Hz), 7.64 (2H, t, *J* = 7.1 Hz), 7.38 (2H, tt, *J* = 7.7, 1.5 Hz), 7.30 (2H, tdd, *J* = 7.5, 4.4, 1.2 Hz), 5.89 (1H, ddt, *J* = 16.5, 11.1, 5.7 Hz), 5.29 (1H, dq, *J* = 17.3, 1.5 Hz), 5.15 (1H, dd, *J* = 10.6, 1.5 Hz), 4.80–4.74 (2H, m), 4.63 (1H, d, *J* = 4.0 Hz), 4.64–4.57 (1H, m), 4.32 (2H, d, *J* = 7.1 Hz), 4.32–4.22 (2H, m), 4.21 (1H, t, *J* = 7.1 Hz), 1.07–0.98 (2H, m), 0.05 (9H, s); ^13^C NMR (Methanol-*d*4, 126 MHz) *δ* 172.4, 171.0, 158.5, 145.2, 145.1, 142.6, 133.1, 128.8, 128.2, 128.2, 126.3, 120.9, 119.0, 72.2, 68.4, 67.2, 65.3, 58.6, 48.2, 18.2, −1.5; HRMS (ESI) calculated for C_27_H_33_O_7_NSiNa^+^ [M + Na]^+^ 534.1920, found 534.1918.

Dipeptide (**S4**): Allyl ester **20** (0.61 g, 1.2 mmol) was added to a 50 mL round-bottom flask equipped with a Teflon stir bar and fitted with a septum under an atmosphere of Ar. DCM (20 mL) was added via syringe and the mixture was stirred until it became homogeneous and colorless. The solution was cooled to 0 °C using an ice bath, and DBU (0.17 mL, 1.2 mmol) was added via syringe. The reaction mixture was stirred for 1.5 h at 0 °C. After the starting material was consumed completely as determined by TLC, HOAt (0.25 g, 1.8 mmol) was added and the mixture was stirred for 5 min at 0 °C using an ice bath. Acid **5** (0.55 g, 1.2 mmol) followed by HATU (0.68 g, 1.8 mmol) were added and the resulting homogeneous yellow reaction mixture was warmed to room temperature and stirred overnight. After 12 h, the reaction mixture was cooled to 0 °C using an ice bath and quenched by the addition of H_2_O (20 mL). The mixture was transferred with H_2_O (20 mL) to a separatory funnel and extracted with EtOAc (2 × 50 mL). The combined organic layers were dried over Na_2_SO_4_. Following the removal of the solvent that provided a white solid, the crude product was purified by column chromatography on silica gel, eluting with EtOAc/hexanes to afford dipeptide **S4** in 60% yield (0.52 g, 0.72 mmol) as a white amorphous solid: R_f_ = 0.54 (ethyl acetate/hexanes = 1:1); ${\left[\alpha \right]}_{D}^{25}$ = −4.70 (*c* 1.0, MeOH); ^1^H NMR (Methanol-*d*4, 500 MHz) *δ* 7.79 (2H, d, *J* = 7.5 Hz), 7.67 (2H, dd, *J* = 15.1, 7.5 Hz), 7.38 (2H, t, *J* = 7.4 Hz), 7.30 (2H, t, *J* = 7.5 Hz), 5.93 (1H, ddt, *J* = 16.5, 11.0, 5.8 Hz), 5.37–5.28 (1H, m), 5.22 (1H, d, *J* = 10.5 Hz), 4.96 (1H, d, *J* = 2.5 Hz), 4.76 (1H, d, *J* = 2.7 Hz), 4.63 (2H, d, *J* = 5.9 Hz), 4.36 (2H, qd, *J* = 10.7, 7.8 Hz), 4.24 (3H, ddt, *J* = 10.9, 7.7, 3.9 Hz), 4.17 (1H, dd, *J* = 8.9, 5.3 Hz), 3.05 (2H, t, *J* = 6.8 Hz), 1.76 (1H, td, *J* = 11.9, 9.8, 5.5 Hz), 1.61 (1H, qd, *J* = 9.5, 6.3 Hz), 1.57–1.48 (2H, m), 1.43 (10H, s), 1.02 (2H, dd, *J* = 9.7, 7.5 Hz), 0.02 (9H, s); ^13^C NMR (Methanol-*d*4, 126 MHz) *δ* 174.9, 172.4, 170.4, 158.5, 158.4, 145.5, 145.1, 142.6, 142.5, 133.2, 128.8, 128.7, 128.2, 126.3, 126.2, 120.9, 119.1, 80.0, 71.9, 68.1, 67.3, 65.3, 56.5, 56.2, 48.4, 40.8, 30.5, 28.8, 27.2, 18.2, −1.6; HRMS (ESI) calculated for C_37_H_51_O_10_N_3_SiNa^+^ [M + Na]^+^ 748.3238, found 748.3236.

Tripeptide (**21**): Under an atmosphere of Ar, dipeptide **S4** (0.78 g, 1.07 mmol) was added to a 50 mL round-bottom flask equipped with a Teflon stir bar and fitted with a septum. DCM (20 mL) was added via syringe and the mixture was stirred until it became homogeneous and colorless. The solution was cooled to 0 °C using an ice bath, and DBU (0.16 mL, 1.07 mmol) was added via syringe. The reaction mixture was stirred for 1.5 h at 0 °C. After the starting material was consumed completely as determined by TLC, HOAt (0.22 g, 1.60 mmol) was added and the mixture was stirred for 5 min at 0 °C using an ice bath. Acid **6** (0.38 g, 1.07 mmol) followed by HATU (0.61 g, 1.60 mmol) were added and the resulting homogeneous yellow reaction mixture was warmed to room temperature. After 12 h, the reaction mixture was cooled to 0 °C using an ice bath and quenched by the addition of H_2_O (5 mL). The mixture was transferred with H_2_O (10 mL) to a separatory funnel and extracted with EtOAc (2 × 40 mL). The combined organic layers were dried over Na_2_SO_4_, filtered, and concentrated. The crude product was purified by column chromatography on silica gel, eluting with EtOAc/hexanes to afford tripeptide **21** in 65% yield (0.59 g, 0.70 mmol) as a white amorphous solid: R_f_ = 0.57 (ethyl acetate/hexanes = 1:1); ${\left[\alpha \right]}_{D}^{25}$ = −7.10 (*c* 1.0, MeOH); ^1^H NMR (Methanol-*d*4, 500 MHz) *δ* 7.79 (2H, d, *J* = 7.5 Hz), 7.68 (2H, dd, *J* = 12.1, 7.5 Hz), 7.39 (2H, t, *J* = 7.5 Hz), 7.31 (2H, t, *J* = 7.4 Hz), 5.94 (1H, ddt, *J* = 16.5, 11.0, 5.8 Hz), 5.34 (1H, dq, *J* = 17.0, 1.5 Hz), 5.24 (1H, dd, *J* = 10.4, 1.5 Hz), 4.97 (1H, d, *J* = 2.8 Hz), 4.76 (1H, d, *J* = 2.8 Hz), 4.62 (2H, t, *J* = 5.4 Hz), 4.56 (1H, s), 4.49 (1H, dd, *J* = 8.7, 5.1 Hz), 4.39 (2H, qd, *J* = 10.6, 6.9 Hz), 4.27–4.18 (3H, m), 4.14 (1H, d, *J* = 5.7 Hz), 3.02 (2H, td, *J* = 6.9, 4.8 Hz), 1.92 (1H, p, *J* = 6.7 Hz), 1.81 (1H, dt, *J* = 14.5, 7.4 Hz), 1.62 (1H, dtd, *J* = 14.3, 9.2, 5.5 Hz), 1.54–1.45 (2H, m), 1.40 (9H, s), 1.20 (1H, tt, *J* = 14.8, 6.5 Hz), 1.01 (2H, t), 0.97–0.84 (6H, m), 0.06 (9H, s); ^13^C NMR (Methanol-*d*4, 126 MHz) *δ* 174.4, 174.0, 172.4, 170.4, 158.9, 158.5, 145.4, 145.1, 142.6, 133.2, 128.8, 128.2, 128.2, 126.3, 120.9, 119.2, 79.9, 72.0, 68.1, 67.3, 65.3, 60.4, 56.5, 54.0, 48.4, 40.8, 38.2, 30.5, 28.8, 27.3, 27.2, 18.2, 15.0, 12.0, −1.6; HRMS (ESI) calculated for C_43_H_62_O_11_N_4_SiNa^+^ [M + Na]^+^ 861.4077, found 861.4076.

Dipeptide (**S5**): Under an atmosphere of Ar, *N*-Boc-*d*-valine (0.36 g, 1.66 mmol) and *d*-leucine allyl ester **22** (0.28 g, 1.66 mmol) were added to a 100 mL round-bottom flask equipped with a Teflon stir bar and fitted with a septum. DMF (30 mL) was added via syringe and the mixture was stirred until it became homogeneous and colorless. The solution was cooled to 0 °C using an ice bath, and DIPEA (1.16 mL, 6.64 mmol) and HOAt (0.34 g, 2.49 mmol) were added and the mixture was stirred for 5 min. HATU (0.95 g, 2.49 mmol) was then added, and the resulting homogeneous reaction solution was warmed to room temperature. After 12 h, the reaction mixture was cooled to 0 °C using an ice bath and quenched by the addition of H_2_O (10 mL). The mixture was transferred with H_2_O (20 mL) to a separatory funnel and extracted with EtOAc (3 × 50 mL). The combined organic layers were washed with aqueous 1 N HCl (2 × 30 mL), brine (2 × 30 mL), H_2_O (30 mL), and brine (30 mL), dried over Na_2_SO_4_, filtered, and concentrated. The crude product was purified by column chromatography on silica gel, eluting with EtOAc/hexanes to afford dipeptide **S5** in 80% yield (0.49 g, 1.33 mmol) as a white amorphous solid: R_f_ = 0.74 (ethyl acetate/hexanes = 1:2); ${\left[\alpha \right]}_{D}^{25}$ = +29.60 (*c* 1.0, MeOH); ^1^H NMR (Chloroform-*d*, 500 MHz) *δ* 6.47 (1H, d, *J* = 7.7 Hz), 5.86 (1H, ddt, *J* = 16.4, 10.9, 5.7 Hz), 5.29 (1H, d, *J* = 17.1 Hz), 5.21 (1H, d, *J* = 10.4 Hz), 5.14 (1H, d, *J* = 9.0 Hz), 4.59 (3H, t, *J* = 7.0 Hz), 3.90 (1H, t, *J* = 7.8 Hz), 2.11–2.02 (1H, m), 1.63 (2H, ddd, *J* = 13.4, 8.5, 5.3 Hz), 1.54 (1H, q, *J* = 8.9 Hz), 1.40 (9H, s), 0.93 (3H, d, *J* = 6.8 Hz), 0.90 (9H, dd, *J* = 5.8, 2.0 Hz); ^13^C NMR (Chloroform-*d*, 126 MHz) *δ* 172.4, 171.7, 155.9, 131.7, 118.6, 79.7, 65.8, 59.9, 50.8, 41.4, 30.9, 28.3, 24.8, 22.8, 21.9, 19.2, 18.0; HRMS (ESI) calculated for C_19_H_34_O_5_N_2_Na^+^ [M + Na]^+^ 393.2361, found 393.2360.

Tripeptide (**23**): Trifluoroacetic acid (5 mL) was added to a solution of compound **S5** (40.42 mg, 2.15 mmol) in dry DCM (10 mL) at 0 °C. The reaction mixture was stirred at room temperature for 2 h. Volatiles were removed in vacuo and the residue was dried under high vacuum for 2 h to afford the corresponding amine, which was used in the next step without any further purification.

DIPEA (1.50 mL, 8.60 mmol) and HOAt (0.44 g, 3.23 mmol) were added to the above amino acid **9** (0.82 g, 2.15 mmol) solution in DCM (30 mL) at 0 °C and the mixture was stirred for 5 min. HATU (1.23 g, 3.23 mmol) was added and the resulting homogeneous reaction mixture was warmed to room temperature. After 12 h, the reaction mixture was cooled to 0 °C using an ice bath and quenched by the addition of H_2_O (10 mL). The mixture was transferred with H_2_O (10 mL) to a separatory funnel and extracted with EtOAc (3 × 50 mL). The combined organic layers were washed with aqueous 1 N HCl (2 × 20 mL), brine (2 × 20 mL), H2O (20 mL), and brine (20 mL), dried over Na_2_SO_4_, filtered, and concentrated. The crude product was purified by column chromatography on silica gel, eluting with EtOAc/hexanes to afford tripeptide **23** in 91% yield over two steps (1.24 g, 1.96 mmol) as a white amorphous solid: R_f_ = 0.40 (ethyl acetate/hexanes = 1:1); ${\left[\alpha \right]}_{D}^{25}$ = +17.10 (*c* 1.0, MeOH); ^1^H NMR (Methanol-*d*4, 500 MHz) *δ* 7.59 (1H, d, *J* = 1.9 Hz), 7.28 (1H, d, *J* = 8.5 Hz), 7.14 (1H, s), 7.04 (1H, dd, *J* = 8.7, 2.0 Hz), 5.91 (1H, ddt, *J* = 17.3, 10.8, 5.6 Hz), 5.31 (1H, dq, *J* = 17.3, 1.7 Hz), 5.25–5.18 (1H, m), 4.57 (2H, d, *J* = 5.7 Hz), 4.49–4.37 (2H, m), 4.15 (1H, t, *J* = 6.0 Hz), 4.13–4.00 (2H, m), 3.18 (1H, dd, *J* = 14.2, 8.3 Hz), 3.04 (1H, dd, *J* = 14.2, 6.9 Hz), 2.09–1.95 (1H, m), 1.81–1.68 (2H, m), 1.60 (1H, td, *J* = 8.4, 3.5 Hz), 0.95 (5H, d, *J* = 6.3 Hz), 0.89 (3H, d, *J* = 6.3 Hz), 0.66 (3H, d, *J* = 6.8 Hz), 0.60 (3H, d, *J* = 6.8 Hz), 0.01 (9H, s); ^13^C NMR (Methanol-*d*4, 126 MHz) *δ* 174.9, 173.7, 173.5, 158.6, 136.4, 133.3, 129.9, 126.5, 125.6, 122.5, 119.0, 118.7, 113.5, 110.7, 66.6, 64.3, 59.7, 57.5, 52.3, 41.1, 31.0, 28.6, 25.8, 23.4, 21.7, 19.4, 18.6, 17.8, −1.5; HRMS (ESI) calculated for C_31_H_47_O_6_N_4_ClSiNa^+^[M + Na]^+^ 657.2845, found 657.2846.

Hexapeptide (**24**): To a solution of allyl ester **23** (202.88 mg, 0.32 mmol) in anhydrous THF (10 mL) was added Pd(PPh_3_)_4_ (18.49 mg, 0.016 mmol) followed by the dropwise addition of morpholine (0.28 mL, 3.2 mmol), and the mixture was stirred for 45 min at room temperature. The reaction mixture was concentrated, and the residue was dissolved in DCM (30 mL). The resulting solution was washed with 1 N HCl (10 mL) and H_2_O (10 mL). The organic layer was dried with anhydrous Na_2_SO_4_ and concentrated. The crude acid intermediate **S7** was obtained as an oil which was used in the next step without further purification.

Tripeptide **21** (268.29 mg, 0.32 mmol) was added to a 25 mL round-bottom flask equipped with a Teflon stir bar and fitted with a septum under an atmosphere of Ar. DCM (10 mL) was added via syringe and the mixture was stirred until it became homogeneous and colorless. The solution was cooled to 0 °C using an ice bath, and DBU (47.81 μL, 0.32 mmol) was added via syringe. The reaction mixture was stirred for 1.5 h at 0 °C. After the starting material was consumed completely as determined by TLC, HOAt (65.33 mg, 0.48 mmol) was added and the mixture was stirred for 5 min at 0 °C using an ice bath. **S7** (190.16 mg, 0.32 mmol) and HATU (182.51 mg, 0.48 mmol) were added and the resulting homogeneous yellow reaction mixture was warmed to room temperature. After 12 h, the reaction mixture was cooled to 0 °C using an ice bath and quenched by the addition of H_2_O (5 mL). The mixture was transferred with H_2_O (5 mL) to a separatory funnel and extracted with EtOAc (2 × 30 mL). The combined organic layers were dried over Na_2_SO_4_, filtered, and concentrated. The crude product was purified by column chromatography on silica gel, eluting with EtOAc/hexanes to afford hexapeptide **24** in 83% yield over two steps (0.32 g, 0.27 mmol) as a white amorphous solid: R_f_ = 0.45 (methanol/dichloromethane = 1:20); ${\left[\alpha \right]}_{D}^{25}$ = −6.30 (*c* 1.0, MeOH:DCM = 1:1); ^1^H NMR (DMSO-*d*6, 400 MHz) *δ* 11.04 (1H, d, *J* = 6.6 Hz), 8.13 (2H, dd, *J* = 9.2, 4.0 Hz), 8.06 (1H, d, *J* = 8.1 Hz), 7.86 (1H, d, *J* = 8.3 Hz), 7.66 (2H, d, *J* = 16.9 Hz), 7.31 (1H, d, *J* = 8.6 Hz), 7.23 (1H, d, *J* = 2.3 Hz), 7.13 (1H, d, *J* = 8.0 Hz), 7.01 (1H, dd, *J* = 8.6, 2.1 Hz), 6.71 (1H, t, *J* = 5.7 Hz), 6.00 (1H, t, *J* = 5.7 Hz), 5.85 (1H, ddt, *J* = 16.2, 10.7, 5.4 Hz), 5.36–5.24 (1H, m), 5.19 (1H, d, *J* = 10.5 Hz), 4.78 (1H, dd, *J* = 9.1, 3.1 Hz), 4.60 (1H, dd, *J* = 6.3, 3.1 Hz), 4.51 (2H, q, *J* = 8.2, 6.8 Hz), 4.35 (4H, dtd, *J* = 17.6, 7.5, 3.3 Hz), 4.13 (3H, q, *J* = 8.7 Hz), 3.91 (2H, dd, *J* = 9.9, 6.7 Hz), 3.01 (1H, dd, *J* = 14.4, 5.2 Hz), 2.86 (3H, q, *J* = 8.4, 7.9 Hz), 1.99 (1H, p, *J* = 6.8 Hz), 1.79 (1H, p, *J* = 6.6 Hz), 1.57 (3H, dq, *J* = 19.5, 13.5, 10.0 Hz), 1.35 (14H, s), 1.08–1.00 (1H, m), 0.94 (2H, t, *J* = 8.5 Hz), 0.85–0.69 (19H, m), 0.00 (9H, s), -0.05 (9H, s); ^13^C NMR (DMSO-*d*6, 101 MHz) *δ* 172.2, 172.0, 171.9, 170.7, 170.5, 169.0, 168.4, 156.1, 155.6, 134.6, 132.2, 128.4, 126.0, 123.0, 120.7, 118.0, 117.9, 112.7, 110.0, 77.4, 70.4, 65.1, 63.2, 61.8, 57.7, 55.4, 54.8, 51.9, 51.5, 42.5, 36.9, 29.9, 29.7, 29.0, 28.3, 27.8, 25.8, 25.8, 24.1, 22.9, 21.4, 19.2, 17.8, 17.3, 16.8, 14.3, 11.6, −1.6; HRMS (ESI) calculated for C_56_H_93_O_14_N_8_ClSi_2_Na^+^ [M + Na]^+^ 1215.5923, found 1215.5931.

Hexapeptide (**S9**): 1.0 M TBAF in THF (0.62 mL, 0.62 mmol) was added to a solution of the hexapeptide **24** (74.0 mg, 0.062 mmol) in THF (10 mL). The reaction mixture was stirred for 1 h at 50 °C. The reaction mixture was then cooled to 0 °C using an ice bath and quenched by the addition of a saturated aqueous solution of NH_4_Cl (5 mL). The mixture was transferred to a separatory funnel and extracted with EtOAc (3 × 30 mL). The combined organic layers were dried over Na_2_SO_4_, filtered, and concentrated. The crude product was used in the next step without further purification.

To a solution of the above crude product in a mixture of DCM (60 mL) and DMF (3 mL) were sequentially added DMAP (1.51 mg, 0.012 mmol), HOAt (12.66 mg, 0.093 mmol) and EDCI (17.83 mg, 0.093 mmol). The reaction mixture was stirred for 48 h at 30 °C. The reaction mixture was then cooled to 0 °C using an ice bath and quenched by the addition of a saturated aqueous solution of NH_4_Cl (20 mL). The mixture was transferred to a separatory funnel and extracted with DCM (3 × 60 mL). The combined organic layers were washed with H_2_O (2 × 30 mL) and brine (30 mL), dried over Na_2_SO_4_, and concentrated. The residue was purified by column chromatography on silica gel, eluting with MeOH/DCM to afford hexapeptide **S9** in 62% yield over two steps (35.77 mg, 0.04 mmol) as a white amorphous solid: R_f_ = 0.30 (methanol/dichloromethane = 1:20); ${\left[\alpha \right]}_{D}^{25}$ = −7.10 (*c* 1.0, MeOH); ^1^H NMR (DMSO-*d*6, 500 MHz) *δ* 11.03 (1H, d, *J* = 2.5 Hz), 8.17 (1H, d, *J* = 6.4 Hz), 8.14 (1H, d, *J* = 7.3 Hz), 7.95 (1H, d, *J* = 7.4 Hz), 7.83 (1H, d, *J* = 8.2 Hz), 7.73 (1H, d, *J* = 8.7 Hz), 7.57 (1H, d, *J* = 2.0 Hz), 7.40 (1H, d, *J* = 7.0 Hz), 7.32 (1H, d, *J* = 8.7 Hz), 7.16 (1H, d, *J* = 2.5 Hz), 7.04 (1H, dd, *J* = 8.6, 2.1 Hz), 6.79 (1H, t, *J* = 5.8 Hz), 5.87 (1H, ddt, *J* = 17.4, 10.7, 5.4 Hz), 5.29 ( 1H, dq, *J* = 17.2, 1.7 Hz), 5.21–5.15 (1H, m), 4.65 (1H, dd, *J* = 8.6, 4.4 Hz), 4.62–4.47 (4H, m), 4.30–4.20 (2H, m), 4.00 (1H, q, *J* = 6.9 Hz), 3.79 (1H, t, *J* = 7.0 Hz), 3.04 (1H, dd, *J* = 14.3, 7.8 Hz), 2.91 (3H, dq, *J* = 17.2, 6.7 Hz), 1.88 (2H, dqt, *J* = 11.0, 6.9, 4.4 Hz), 1.81–1.71 (1H, m), 1.62–1.52 (3H, m), 1.48 (1H, dt, *J* = 13.4, 6.8 Hz), 1.36 (10H, s), 1.30 (2H, dt, *J* = 11.1, 5.8 Hz), 1.14 (1H, dp, *J* = 14.6, 7.4 Hz), 0.89–0.80 (12H, m), 0.63 (6H, t, *J* = 7.5 Hz); ^13^C NMR (DMSO-*d*6, 126 MHz) *δ* 172.0, 171.3, 171.2, 171.2, 171.1, 170.5, 168.7, 155.6, 134.6, 132.4, 128.4, 125.6, 123.0, 120.8, 117.8, 117.6, 112.8, 109.6, 77.4, 70.1, 64.9, 59.7, 56.5, 56.0, 55.3, 54.0, 53.2, 51.7, 36.3, 29.2, 28.3, 27.5, 26.8, 26.2, 25.7, 24.3, 22.7, 22.1, 18.8, 18.5, 18.0, 14.5, 11.4; HRMS (ESI) calculated for C_45_H_67_O_11_N_8_ClNa^+^ [M + Na]^+^ 953.4507, found 953.4510.

Nicrophorusamide A (**25**): Cyclic hexapeptide **S9** (46.52 mg, 0.05 mmol) was dissolved in a saturated methanolic NH_3_ solution (10 mL) and stirred at 0 °C. After the starting material was consumed completely as determined by TLC, the reaction was concentrated by rotary evaporation. The residue was azeotropically dried by concentration from methanol (2 × 10 mL) and then used in the next step without further purification.

To a solution of the above amide in dry DCM (5 mL) was added trifluoroacetic acid (2 mL) at 0 °C. The reaction mixture was stirred at ambient temperature for 0.5 h. Volatiles were removed in vacuo and the residue was purified by column chromatography on silica gel, eluting with MeOH/DCM to afford **25** in 55% yield (21.71 mg, 0.0275 mmol) as a white amorphous solid: R_f_ = 0.27 (methanol/dichloromethane = 1:2); ${\left[\alpha \right]}_{D}^{25}$ = −15.50 (*c* 0.2, MeOH); IR (MeOH) 3441, 3385, 2964, 1676, 1631, 1535, 1201, 1055 cm^−1^; ^1^H NMR (DMSO-*d*6, 500 MHz) *δ* 11.05 (d, *J* = 2.4 Hz, 1H), 8.48 (d, *J* = 6.0 Hz, 1H), 7.97 (d, *J* = 7.9 Hz, 1H), 7.85 (dd, *J* = 7.5, 4.8 Hz, 2H), 7.76 (d, *J* = 6.1 Hz, 3H), 7.60–7.55 (m, 2H), 7.48 (d, *J* = 7.3 Hz, 1H), 7.33 (d, *J* = 8.6 Hz, 2H), 7.28 (d, *J* = 2.7 Hz, 1H), 7.15 (d, *J* = 2.3 Hz, 1H), 7.04 (dd, *J* = 8.5, 2.1 Hz, 1H), 5.91 (d, *J* = 6.0 Hz, 1H), 4.65–4.56 (m, 2H), 4.47 (d, *J* = 3.8 Hz, 1H), 4.24 (ddt, *J* = 9.7, 7.3, 4.0 Hz, 2H), 3.99 (q, *J* = 6.7 Hz, 1H), 3.75 (t, *J* = 7.3 Hz, 1H), 3.11 (dd, *J* = 14.2, 7.3 Hz, 1H), 2.89 (dd, *J* = 14.2, 7.0 Hz, 1H), 2.81 (t, *J* = 6.6 Hz, 2H), 1.86 (dh, *J* = 28.2, 6.8 Hz, 3H), 1.70–1.59 (m, 1H), 1.58–1.54 (m, 2H), 1.54–1.48 (m, 2H), 1.34 (dq, *J* = 13.8, 6.8 Hz, 1H), 1.15 (dp, *J* = 14.8, 7.4 Hz, 1H), 0.89–0.75 (m, 13H), 0.70 (dd, *J* = 10.9, 6.7 Hz, 6H); ^13^C NMR (DMSO-*d*6, 126 MHz) *δ* 173.2, 172.1, 171.7, 171.1, 170.6, 170.4, 169.5, 134.5, 128.4, 125.5, 123.0, 120.7, 117.5, 112.7, 109.8, 70.8, 60.4, 56.6, 55.7, 54.8, 53.5, 53.2, 51.5, 38.4, 36.2, 29.1, 26.8, 26.7, 25.6, 24.3, 23.8, 22.7, 21.8, 18.8, 18.4, 14.5, 11.3; HRMS (ESI) calculated for C_37_H_57_O_8_N_9_Cl^+^ [M + H]^+^ 790.4009, found 790.4013.

## 4. Conclusions

In summary, the first total synthesis of nicrophorusamide A proceeds in 10 longest linear steps from the known acid **19** with an overall yield of 6.3%. The enantioselective synthetic strategy with high stereochemical fidelity also led to the completion of the synthesis of noursamycin A in 5.5% overall yield from acid **10**, which suggests that the reported structure of noursamycin A has been assigned incorrectly. The study disclosed herein highlighted the importance of synthetic endeavors in structural elucidations of natural products.

## Data Availability

Data are contained within the article and Appendix A.

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
