# Peer review of "Total Synthesis of Nicrophorusamide A and Structural Disproof of the Proposed Noursamycin A"

_molecules, 2023, doi:10.3390/molecules28217442_

Round 1

Reviewer 1 Report

Comments and Suggestions for Authors

“Total synthesis of nicrophorusamide A and structural disproof of the proposed noursamycin A” by Ye et al. describes the total synthesis of these two cyclic polypeptides, highlighting an inconsistency with the structure reported in the literature for noursamycin A and therefore correcting its stereochemistry.

The authors adopted a simple and rapid approach using different condensation reagents and protecting groups for the construction of these cyclic peptides, obtaining both products in moderate yields. Although the synthesis does not seem particularly innovative and the yield is not surprising, the idea seems intriguing and good; the overall synthesis is streamlined and direct and the characterization of the compounds is in-depth and complete.

I found the manuscript to be well structured, with solid and consistent data to support all conclusions reached. It is a well written and easy to understand document.

Therefore, I recommend publishing on Molecules after the following minor corrections:

- in the introduction chapter, the biological activity of these two hexapeptides should be slightly expanded

- Scheme 3 seems slightly deformed compared to the other figures; please correct

- Line 75: Provide Teoc-Osu full name

- Line 88: correct “TABF” with “TBAF”

- The final section should be slightly expanded, commenting on the importance of correctly identifying the structure of natural compounds using the work done as an example.

Author Response

See attached file>

- in the introduction chapter, the biological activity of these two hexapeptides should be slightly expanded

Reply: Revised accordingly

- Scheme 3 seems slightly deformed compared to the other figures; please correct

Reply: Revised.

- Line 75: Provide Teoc-Osu full name

Revised.

- Line 88: correct “TABF” with “TBAF”

Revised.

- The final section should be slightly expanded, commenting on the importance of correctly identifying the structure of natural compounds using the work done as an example.

Reply: Revised accordingly

Reviewer 2 Report

Comments and Suggestions for Authors

The authors report on the total synthesis of two cyclopeptides, nicrophorusamide A and noursamycin A. In my opinion this paper is not suitable for publication.

First, all the references I checked are not fitting, including the references regarding the isolation and elucidation of the structure of the two cyclopeptides, which are the subject of the manuscript (references 5 and 6), self-citations (references 3 and 4), and others (see for example, refs 1, 2, 8 etc…).

Main point: the authors state that “The total synthesis of the proposed noursamycin A was accomplished, which disproved the original stereochemical assignments”, but they only observe that the NMR spectroscopic data of their synthetic peptide “with those reported for natural noursamycin A revealed that they were similar, but not identical”. This does not prove a difference in stereochemical assignment, nor does it suggest what the difference is between the two structures. Furthermore, the comparison appears to be based essentially on 13C NMR spectra (see SI). Unfortunately, in the original article the 13C NMR spectrum of the natural peptide was not reported but was deduced from two-dimensional NMR spectra. This fact could explain small differences in the values. In addition, a different concentration of the NMR sample, which is not reported, could also lead to variations in the resonance values.

Reviewer 3 Report

Comments and Suggestions for Authors

In this manuscript the authors have completed the total synthesis of the proposed structure of natural product noursamycin A. Because the NMR spectroscopic data of the synthetic natural product did not match with those reported for the isolated natural product, the authors employed the same synthetic strategy to access a structurally close natural compound named nicrophorusamide A. The confirmation that the spectroscopic data of synthetic nicrophorusamide A and naturally isolated match suggest that the reported structure of noursamycin A was assigned incorrectly.

From the chemistry point of view, despite the authors used standard peptide synthesis and did not bring novelty through the synthetic strategy, it is effective to disproff the proposed structure of noursamycin A, highlighting the importance of total synthesis in structural elucidation of natural products. In general, the reviewer consider the manuscript is a nice work and it is of interest for the readers of Molecules and should be published in Molecules after minor revisions.

Specific points:

In the procedure and analytical description of compounds, there are some of them that are not in the schemes: 19-1, 20-1, 22-1, 24-1. In order to have a better following of the compounds, I would suggest to the authors to include the transformations where these compounds are involved in the SI and re-named them as S1, S2, etc. It will be easier to follow by the reader.

Figure S1 shows the comparison of 13C spectra for synthetic and natural nicrophorusamide A. It will be of great interest to include also the comparison of 1H NMR spectra, as well as 1H and 13C NMR spectra comparison for noursamycin A.

In the procedure and analytical description of compounds, there are some compounds numbers that are not in bold: line 151 acid 10,  line 218, acid 6, line 272 compound 15, etc. The authors should carefully check this section.

Author Response

See attached file, please.

Comments and Suggestions for Authors

In this manuscript the authors have completed the total synthesis of the proposed structure of natural product noursamycin A. Because the NMR spectroscopic data of the synthetic natural product did not match with those reported for the isolated natural product, the authors employed the same synthetic strategy to access a structurally close natural compound named nicrophorusamide A. The confirmation that the spectroscopic data of synthetic nicrophorusamide A and naturally isolated match suggest that the reported structure of noursamycin A was assigned incorrectly.

From the chemistry point of view, despite the authors used standard peptide synthesis and did not bring novelty through the synthetic strategy, it is effective to disproff the proposed structure of noursamycin A, highlighting the importance of total synthesis in structural elucidation of natural products. In general, the reviewer consider the manuscript is a nice work and it is of interest for the readers of Molecules and should be published in Molecules after minor revisions.

Specific points:

In the procedure and analytical description of compounds, there are some of them that are not in the schemes: 19-1, 20-1, 22-1, 24-1. In order to have a better following of the compounds, I would suggest to the authors to include the transformations where these compounds are involved in the SI and re-named them as S1, S2, etc. It will be easier to follow by the reader.

Revised. The transformation of these compounds was added to supplementary information and renamed S1, S2, etc.

Figure S1 shows the comparison of 13C spectra for synthetic and natural nicrophorusamide A. It will be of great interest to include also the comparison of 1H NMR spectra, as well as 1H and 13C NMR spectra comparison for noursamycin A.

Revised. The comparison of 1H NMR spectra for synthetic and natural nicrophorusamide A, as well as the comparison of 1H and 13C NMR spectra for noursamycin A, was added to the supplementary information.

In the procedure and analytical description of compounds, there are some compounds numbers that are not in bold: line 151 acid 10,  line 218, acid 6, line 272 compound 15, etc. The authors should carefully check this section.

Revised.

Round 2

Reviewer 2 Report

Comments and Suggestions for Authors

Abstract, line 11: replace stereochemical with structural

line 29: replace the term  'confounding'